# The viral packaging motor potentiates Kaposi's sarcoma-associated herpesvirus gene expression late in infection

Chloe O. McCollum[1], Allison L. Didychuk [2¤a]*, Dawei Liu[3], Laura A. Murray-Nerger[3¤b¤c], Ileana M. Cristea[3], Britt A. Glaunsinger [1,2,4]*

**1** Department of Molecular and Cell Biology, University of California Berkeley, Berkeley, California, United States of America, **2** Department of Plant and Microbial Biology, University of California Berkeley, Berkeley, California, United States of America, **3** Department of Molecular Biology, Princeton University, Princeton, New Jersey, United States of America, **4** Howard Hughes Medical Institute, University of California Berkeley, Berkeley, California, United States of America

¤a Current address: Department of Molecular Biophysics & Biochemistry, Yale University, New Haven, Connecticut, United States of America
¤b Current address: Division of Infectious Diseases, Department of Medicine, Brigham and Women's Hospital, Boston, Massachusetts, United States of America
¤c Current address: Department of Microbiology, Harvard Medical School, Boston, Massachusetts, United States of America
* allison.didychuk@yale.edu (ALD); glaunsinger@berkeley.edu (BAG)

**Data Availability Statement:** All mass spectrometry data have been deposited on the Pride proteome exchange public repository. Project

## Abstract

β- and γ-herpesviruses transcribe their late genes in a manner distinct from host transcription. This process is directed by a complex of viral transcriptional activator proteins that hijack cellular RNA polymerase II and an unknown set of additional factors. We employed proximity labeling coupled with mass spectrometry, followed by CRISPR and siRNA screening to identify proteins functionally associated with the Kaposi's sarcoma-associated herpesvirus (KSHV) late gene transcriptional complex. These data revealed that the catalytic subunit of the viral DNA packaging motor, ORF29, is both dynamically associated with the viral transcriptional activator complex and potentiates gene expression late in infection. Through genetic mutation and deletion of ORF29, we establish that its catalytic activity potentiates viral transcription and is required for robust accumulation of essential late proteins during infection. Thus, we propose an expanded role for ORF29 that encompasses its established function in viral packaging and its newly discovered contributions to viral transcription and late gene expression in KSHV.

## Author summary

β- and γ-herpesviruses express a class of genes essential for completion of the viral life cycle late during infection. A specialized complex of viral transcriptional activator proteins drives expression of these late genes in a manner dependent on viral genome replication, although the mechanisms and regulation of this process are largely unknown. Here, we identified factors that physically and functionally associate with the late gene

Name: KSHV ORF18 TurboID in human iSLK cell lines Project accession: PXD039694 Project DOI: 10.6019/PXD039694.

**Funding:** This work was funded by National Institutes of Health grant AI122528 to BAG. BAG is an investigator of the Howard Hughes Medical Institute. ALD was supported by a Rhee Family Fellowship of the Damon Runyon Cancer Research Foundation (DRG-2349-18). This work was additionally funded by National Institutes of Health grant GM114141 to IMC. LAM was supported by National Science Foundation Graduate Research Fellowship DGE-1656466 and National Institutes of Health NIGMS T32GM007388. The funders had no role in study design, data collection and analysis, decision to publish, or preparation of the manuscript.

**Competing interests:** The authors have declared that no competing interests exist.

transcription complex and unexpectedly found that the viral DNA packaging motor in Kaposi's sarcoma-associated herpesvirus (KSHV) contributes to gene expression late in infection. We show that the catalytic activity of this protein is not only required for genomic packaging but also for the robust expression of late genes to ensure the successful production of progeny virions. Thus, gene transcription late in infection is mechanistically linked to the conserved processes of viral genome replication and packaging.

## Introduction

The viral gene expression cascade for all double stranded DNA viruses is temporally divided by the process of viral DNA replication. Expression of early genes, which are generally involved in transcription and replication, initiates prior to viral DNA replication, whereas transcription of late genes, which are generally involved in virion morphogenesis, occurs only on newly replicated viral DNA. The replicated genomes thus serve as substrates for transcription and for DNA packaging into capsids. Packaging is driven by an ATP-dependent motor complex called the terminase, which feeds the DNA into preformed icosahedral capsids and cleaves the packaged unit-length genome from the unpackaged concatemeric genome [1].

Early genes have promoters whose structure and preinitiation complex (PIC) assembly are thought to largely resemble their cellular counterparts. In contrast, for the β- and γ-herpesviruses like cytomegalovirus (CMV) and Kaposi's sarcoma-associated herpesvirus (KSHV), late genes have strikingly minimal promoters, consisting only of a TATT sequence followed by a short degenerate motif in lieu of a canonical TATA box [2,3]. Late gene transcription additionally requires a dedicated set of six essential viral transcriptional activator (vTA) proteins that form a complex at late gene promoters [4–10]. In KSHV, the vTAs are encoded by ORF18, ORF24, ORF30, ORF31, ORF34, and ORF66. The best understood of these factors is ORF24, a unique transcriptional coordinator that acts as a viral mimic of cellular TATA-binding protein (TBP) and directly binds both late gene promoters and RNA polymerase II [5,11–13]. The other vTAs contact one another extensively within the complex, forming interactions that are critical for late gene transcription [6–8,14–16]. Several additional factors in β- and γ-herpesviruses have been implicated in late gene expression. These proteins are thought to either contribute directly to this process, influencing mRNA transcription or export, or indirectly through their role in DNA replication, which is required for late gene transcription [17–20].

Late gene production is essential for production of progeny virions [5,8,14,21,22], yet the principles that define the broader regulation of late gene expression remain elusive. We sought to address this gap by identifying viral and cellular factors functionally associated with the vTA complex during late gene transcription. Given that transcription is a highly dynamic process, we used a biotin-based proximity labeling-mass spectrometry approach coupled with functional screens to identify new regulators of late gene expression. Surprisingly, we captured no cellular general transcription factors. However, we discovered an unexpected role for the catalytic subunit of the viral terminase, ORF29. ORF29 is canonically involved in viral genome packaging, yet its catalytic activity was also required to potentiate KSHV gene expression late in infection. Thus, late gene expression is not only licensed by viral DNA replication machinery, but is also supported by the viral DNA packaging motor, suggesting that the three processes may be intertwined.

## Results

### Identification of cellular and viral factors associated with the KSHV late gene transcription complex using proximity labeling-mass spectrometry

To identify viral and host factors involved in regulation of gammaherpesviral late gene expression, we sought to identify proteins associated with the vTA complex during the period of infection when late gene transcription occurs. Because transcription complexes are highly dynamic, we used a proximity labeling system involving the promiscuous biotin ligase TurboID. TurboID can label surrounding proteins in as little as 10 minutes, making it well suited for capturing a snapshot of late gene expression with high temporal sensitivity [23]. To identify which vTA was the best candidate for tethering to the ligase, we first transiently transfected recombinant C-terminal vTA-TurboID-3xHA fusion constructs into HEK293T cells and observed that ORF18, ORF30, ORF31, ORF34, and ORF66 fusion proteins all tolerated addition of the ligase and expressed well (S1A Fig). To test whether fusion of TurboID impaired vTA function, we used a previously described reporter assay to measure late gene transcription [7]. In this assay, the firefly luciferase gene is under control of either an early gene (ORF57) or late gene (K8.1) promoter. The vTA TurboID-fusion proteins were individually transfected into HEK293T cells along with the five other vTA complex members, firefly luciferase reporter, and a renilla luciferase reporter as a control for transfection efficiency. ORF18-TurboID and ORF30-TurboID had comparable late gene expression to the WT viral pre-initiation complex, whereas fusion of TurboID to ORF31, ORF34 and ORF66 reduced their activity (S1B Fig).

The coding density of the viral genome makes the generation of endogenous fusion proteins challenging. While the C-terminus of ORF30 overlaps extensively with neighboring ORF31, the C-terminus of ORF18 is relatively free. Therefore, we generated a recombinant KSHV bacterial artificial chromosome 16 (BAC16) that expressed ORF18 fused at the C-terminus to TurboID-3xHA (Fig 1A). Recombinant virus was generated using the Red recombinase system [24] (S1C Fig). The sequence of the ORF18-TurboID BAC was confirmed by full plasmid sequencing, and restriction enzyme digest indicated that no gross recombination events had occurred during mutagenesis (S1D Fig). The recombinant ORF18-TurboID BAC was transfected into HEK293T cells followed by co-culture with the iSLK renal carcinoma cell line to generate latently infected iSLK cells. These cells harbor the major lytic transactivator, ORF50 (RTA) under control of a doxycycline-inducible promoter, which allows for efficient reactivation from latency [24].

ORF18 is an essential gene [21], and thus to confirm that the fusion did not disrupt its function in KSHV we first assessed progeny virion production for the ORF18-TurboID virus by supernatant transfer assay. The BAC16 virus constitutively expresses GFP, which allows for quantification of infected cells by flow cytometry. The ORF18-TurboID virus infected naive cells at a level comparable to the untagged WT (Fig 1B) and expressed similar levels of representative KSHV early and late proteins (S1E Fig), confirming that the endogenously tagged version of ORF18 is functional.

We next carried out proximity labeling in ORF18-TurboID cells to identify proteins associated with the vTA complex. Robust late gene transcription occurs by 48 h post reactivation and is maintained through the end of the infectious cycle. Therefore, reactivated cells were supplemented with biotin at 48 h post reactivation to covalently biotinylate proteins in the vicinity of the vTA complex (ORF18-TurboID + biotin). As ORF18 localizes predominantly to the nucleus [14], we used WT iSLKs transduced with TurboID bearing a nuclear localization sequence (TurboID-NLS) as a negative control. Reactivated ORF18-TurboID cells not supplied with biotin (ORF18-TurboID, no biotin) and WT iSLKs supplemented with biotin

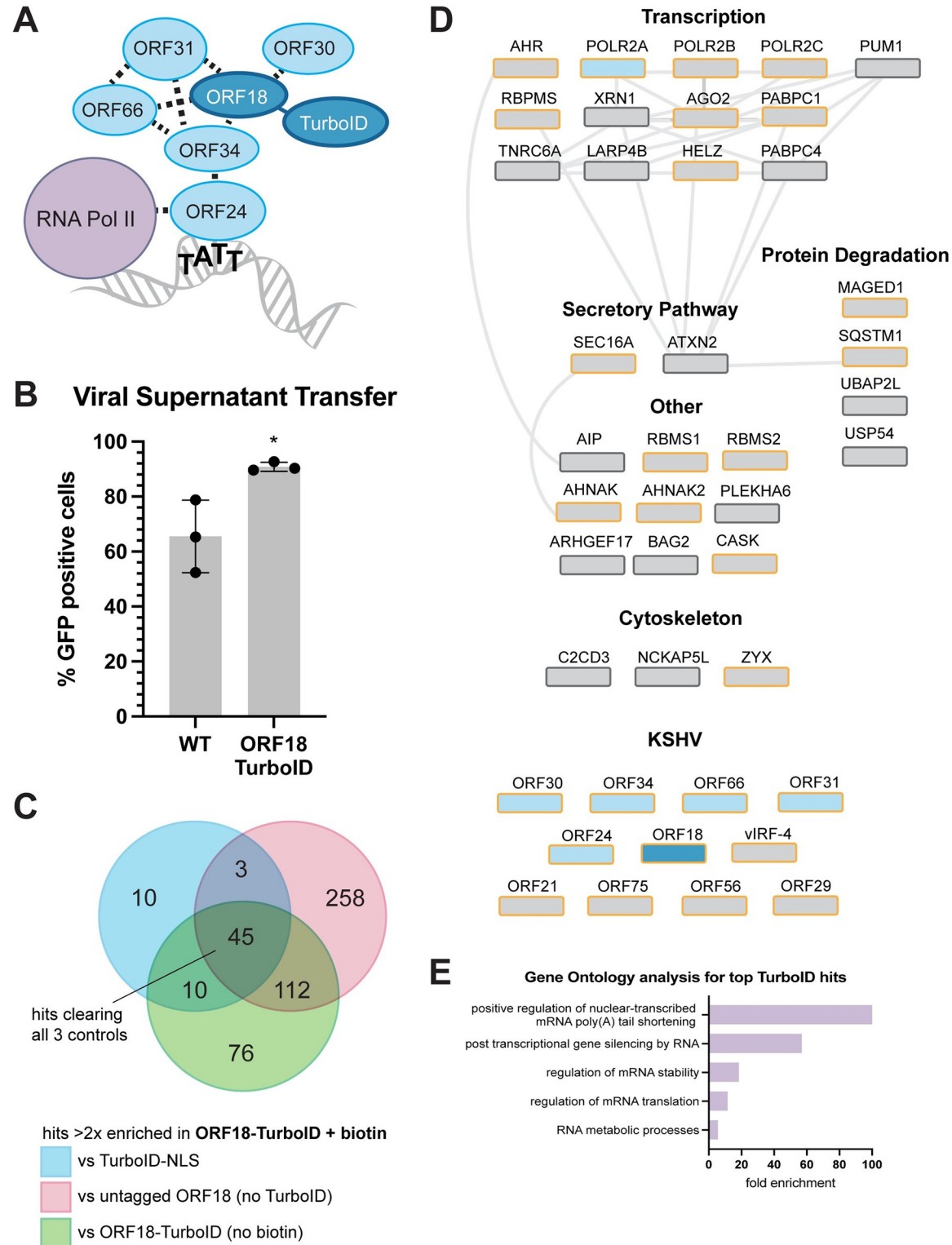

**Fig 1. The vTA complex is associated with numerous cellular and viral factors during infection.** (A) TurboID was tethered to the C-terminus of ORF18. ORF18 directly contacts ORF30, ORF31, ORF66, and ORF34 and is proximal to ORF24 and RNA polymerase II. (B) Infectious virion production from reactivated iSLKs was measured by viral supernatant transfer onto HEK293T cells using flow cytometry for the GFP-expressing virus. Data are from three independent biological replicates and statistics are calculated using an unpaired *t* test. * = P<0.05. (C) Overlap between proteins identified by mass spectrometry with ≥2 unique peptides that demonstrated

>2-fold increase in the ORF18-TurboID + biotin test condition compared to the three negative controls, across 2 biological replicates. Forty-five high confidence hits were specifically enriched in ORF18-TurboID + biotin sample using these filtering criteria. (D) STRING protein-protein interaction network of high-confidence cellular proteins (above) and KSHV proteins (below). ORF18 is shown in dark blue. Known interactors of ORF18 are in light blue, and all other proteins are shown in grey. Proteins with nuclear localization annotated in UniProt are outlined in gold. (E) Gene ontology enrichment analysis of cellular ORF18-TurboID hits. All major enriched functional classes are shown.

(untagged ORF18, no TurboID) served as additional negative controls. Cells were biotinylated for 10 minutes before the labeling reaction was quenched and cells were harvested for analysis.

Biotinylated proteins were enriched by affinity purification with magnetic streptavidin beads and prepared for mass spectrometry. Forty-five proteins demonstrated at least 2-fold enrichment in the ORF18-TurboID + biotin condition over all three negative controls across both replicates and were thus considered candidate vTA interactors (Fig 1C, complete dataset S1 Table). A STRING protein-protein interaction network analysis of the cellular hits suggested several hubs of physical and functional association, with the largest group consisting of proteins associated with transcription (Fig 1D). Gene ontology term analysis [25–27] revealed strong associations with mRNA processing, mRNA stability, and RNA-mediated gene silencing among the high confidence hits (Fig 1E). The viral factors identified are also known to participate in temporally and spatially linked processes including viral DNA replication, late gene transcription, and genomic packaging. Notably, the ORF18-proximal proteome included all known ORF18-interacting proteins, providing support for the validity of this dataset.

## Functional analysis of ORF18-proximal proteins identifies potential late gene expression regulators

To identify which of the candidate ORF18-interacting proteins are functionally important for late gene expression, we carried out a CRISPR-based knockout screen using iSLK cells stably expressing Cas9. These cells contained a version of KSHV containing an eGFP reporter driven by the K8.1 late gene promoter to read out on late gene expression, together with a mIFP reporter constitutively expressed in all cells carrying the viral episome (Fig 2A). Cas9 was stably introduced to iSLK cells carrying the reporter virus. Each candidate interactor identified in the MS dataset was targeted using a pool of three separate guides per gene. Stable CRISPR knockout lines were generated for all candidates with the exceptions of *POLR2A*, *POLR2B*, and *POLR2C*, which encode RNA polymerase II subunits and failed to survive selection. Guides targeting the cellular and viral genomes in regions without expected function (cSAFE and vSAFE respectively) were included as controls along with non-targeting (NT) guides that lacked predicted binding sites on either genome.

The CRISPR knock out late gene reporter cells were reactivated with doxycycline and sodium butyrate, and the proportion of late gene-expressing GFP-positive cells was quantified by flow cytometry for each viral (Fig 2B) and cellular (Fig 2C) knockout 72 h post-reactivation. Among the viral proteins, all 6 vTA knockouts demonstrated expected defects in late gene expression. The loss of late gene expression seen for the ORF56 knockout cells was also anticipated as ORF56 is the KSHV primase for viral DNA replication [28], and DNA replication is required for late gene transcription [29]. Unexpectedly, the ORF29 knockout cells had a decrease in late gene reporter signal that was comparable to the ORF18 knockout. ORF29, also known as TRM3, is proposed to be a component of the KSHV terminase complex responsible for cleavage and packaging of viral DNA into nascent capsids [30]. While packaging and late gene transcription are thought to occur concurrently, there is no evidence of a more direct relationship between the two processes in herpesviruses. ORF75 is a tegument protein required for lytic replication of KSHV [31] that also had a significantly reduced level of late gene

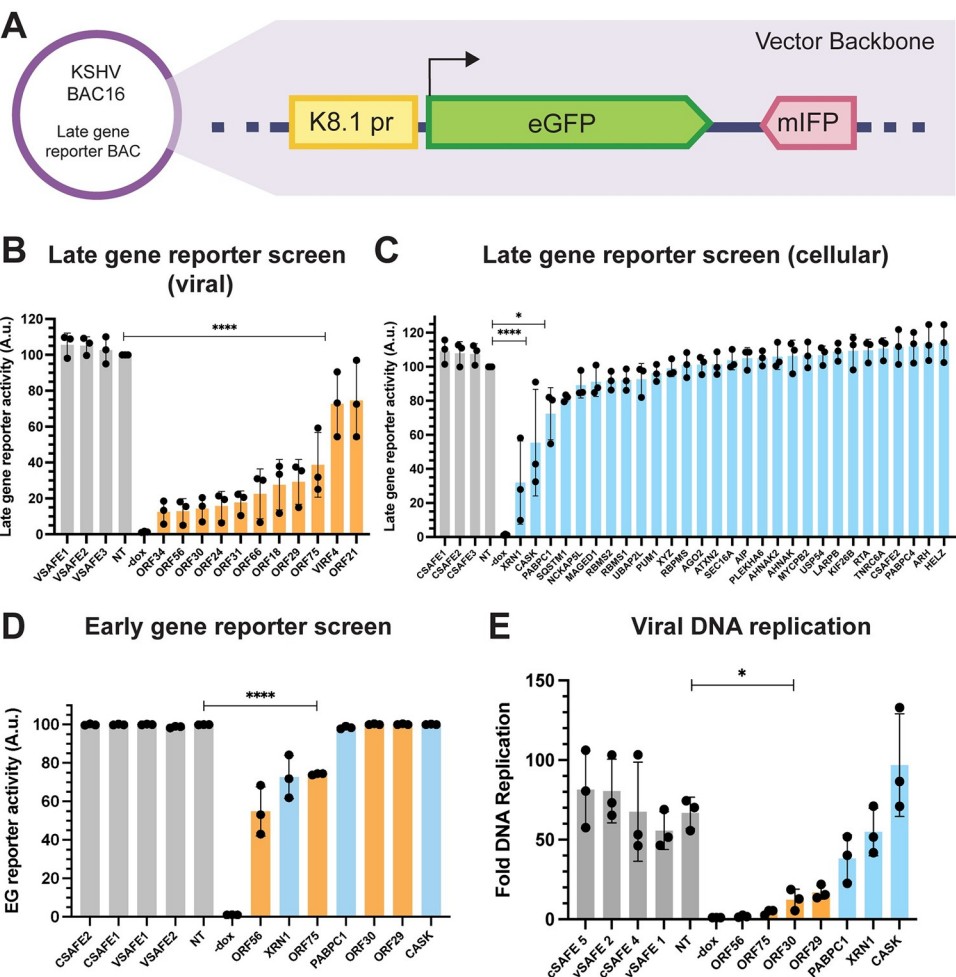

**Fig 2. Functional analysis of ORF18-interacting proteins.** (A) Schematic of the late gene reporter virus used to measure late gene expression by flow cytometry. The reporter construct was cloned into a region of the vector backbone of the KSHV BAC16. The K8.1 late gene promoter drives expression of eGFP, and mIFP is constitutively expressed from the EF-1a promoter. (B) Late gene reporter signal for CRISPR knockouts of viral genes at 72 h post reactivation. Knockouts were normalized against the non-targeting control (NT) within each replicate. Data are from three independent biological replicates, **** = P<0.0001. P values calculated from ordinary one-way ANOVA test. (C) Late gene reporter signal for CRISPR knockouts of cellular genes at 72 h post reactivation. Knockouts were normalized against the non-targeting control (NT) within each replicate. Data are from three independent biological replicates, **** = P<0.0001, * = P<0.05. P values calculated from ordinary one-way ANOVA test. (D) Early gene reporter signal for CRISPR knockouts of viral and cellular genes at 48 h post reactivation. Knockouts were normalized against the non-targeting control (NT) within each replicate. Data are from three independent biological replicates, **** = P<0.0001. P values calculated from ordinary one-way ANOVA test. (E) Viral DNA replication was measured using qPCR to compare reactivated to non-reactivated cells for each CRISPR knockout line. Data are from three independent biological replicates, * = P<0.05. P values calculated from ordinary one-way ANOVA test.

expression. The other two viral proteins identified in the MS dataset, vIRF4 and ORF21, did not differ significantly from the viral safe targeting controls and were not pursued further.

Unlike the viral hits, most of the cellular genes targeted by CRISPR knockout did not affect late gene expression by the reporter assay. However, knockouts of *XRN1*, *CASK*, and *PABPC1* had significantly reduced levels of late gene expression compared to the non-targeting controls.

We next aimed to determine whether the knockout hits from the CRISPR screen affected late genes specifically, or if they were more broadly required for viral gene expression. To

address this, we performed a series of additional CRISPR knockouts using a different reporter virus that has eGFP under control of the ORF57 early gene promoter (S2A Fig). As the six vTAs identified in the MS screen are already known to be selectively required for late gene transcription, only one vTA, ORF30, was included as a control in this early gene reporter screen. The proportion of early gene-expressing GFP-positive cells was quantified by flow cytometry 48 h post reactivation (Fig 2D). Knockouts of ORF56, ORF75 and XRN1 had reduced levels of early gene expression, while all other knockouts (ORF30, ORF29, *CASK* and *PABPC1*) were unimpaired, suggesting they may have a specific effect on late gene expression.

Finally, we tested whether the loss of any proteins impacted viral DNA replication, as this process is required for late gene transcription (Fig 2E). As anticipated, knockout of the viral primase ORF56 caused a total loss of DNA replication, as did knockout of ORF75. Loss of the vTA ORF30 caused a ~6-fold defect in viral DNA replication, in accordance with the replication defects we have previously observed when knocking out other vTAs [2,8]. CRISPR knockout of ORF29 caused a DNA replication defect comparable to that caused by the loss of ORF30, while knockouts for all cellular genes were largely similar to the controls. Taken together, these results suggest that of the many cellular and viral factors that are associated with the vTAs during infection, several may contribute specifically to late gene expression.

## PABPC1 and PABPC4 impact lytic reactivation

Next, we aimed to further probe the contributions to late gene expression of the three cellular genes that emerged from the CRIPSR reporter screen. We used siRNAs to individually knock down the genes of interest in WT iSLKs, then reactivated the cells and monitored expression of several late genes by western blot (Fig 3A). Surprisingly, although knockout of these genes by Cas9 resulted in modestly reduced levels of the late gene reporter in the CRISPR screen, we did not observe any reductions in late gene protein levels with the siRNA knockdowns. One of these proteins, PABPC1, is a poly(A) binding protein that shuttles into the nucleus during

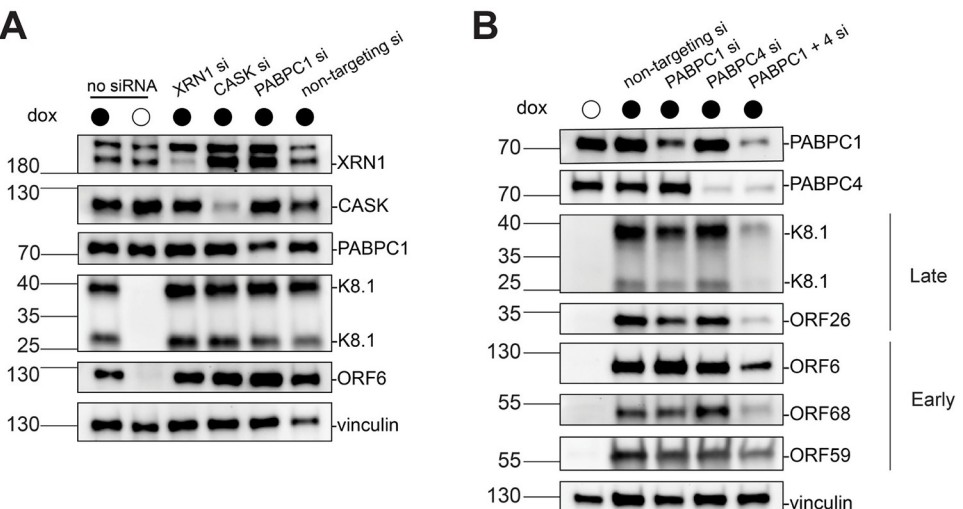

**Fig 3. siRNA-based depletion of select TurboID hits suggests a generalized role for PABC proteins in viral gene expression.** (A) Representative western blots of siRNA-treated WT iSLK cell lysates at 48 h post reactivation showing the efficiency of XRN1, CASK, and PABPC1 depletion as well as levels of the representative late protein, K8.1, and early protein, ORF6. Vinculin serves as a loading control. (B) Representative western blots of WT iSLK cells at 48 h post reactivation that were depleted of PABPC1 and PABPC4 individually or in tandem by siRNA treatment. Levels of late proteins K8.1 and ORF26 are shown, as are levels of early proteins ORF6, ORF68, and ORF59. Vinculin serves as a loading control.

infection [32]. It does so with many other proteins including PABPC4, which also appeared in the MS dataset, but did not show any late gene expression defect in the CRISPR screen reporter assay. Given that depletion of PABPC1 can result in compensatory induction of PABPC4, and that these proteins can function redundantly [33], we co-depleted PABPC1 and PABPC4 (Fig 3B). Under these conditions we saw a pronounced reduction of late gene protein levels. However, we also observed reduced levels of early gene proteins, suggesting that the co-depletion of PABPC1 and PABPC4 may impact lytic gene expression more generally.

## Loss of ORF29 impairs late gene expression

ORF29 was the only viral protein apart from the vTAs that contributed specifically to late gene expression in the CRISPR reporter screens. Although presumed to function analogously to its other herpesviral terminase homologs, ORF29 has not been characterized in KSHV. We therefore generated an ORF29 rabbit polyclonal antibody, which revealed by western blot that it is expressed by 36 h post reactivation in iSLK cells (Fig 4A). The appearance of ORF29 protein preceded that of the true late protein, K8.1, which appeared robustly by 48 h post reactivation, and followed the expression of early proteins ORF59, ORF6, and ORF68, which were observable by 24 h post reactivation. Thus, ORF29 has expression kinetics that place it between that of early and true late genes.

Next, we generated a recombinant ORF29-deficient virus (ORF29.stop) to probe its contribution to late gene expression in an orthogonal manner to the CRISPR screen. A corresponding mutant rescue ORF29 virus (ORF29.MR) was engineered to confirm that any phenotypes observed were due to the loss of ORF29 specifically and not because of additional mutations elsewhere in the BAC generated during mutagenesis (S3A Fig). The sequences of these BACs were confirmed by Sanger sequencing and BAC integrity was assessed by restriction enzyme digest (S3B Fig). ORF29 is conserved across α-, β-, and γ-herpesviruses, and its homologs have been shown to be essential for completion of the viral lifecycle [22,34]. As expected for a terminase mutant, the ORF29.stop virus was incapable of generating infectious progeny virions by supernatant transfer assay, whereas the ORF29.MR virus infected naive cells comparably to WT (Fig 4B).

We next tested the expression levels of several representative early and late proteins by western blot in reactivated cells. Compared to WT and ORF29.MR, the ORF29.stop virus had reduced levels of the late proteins ORF26 and K8.1, while the early proteins ORF6, ORF59, and ORF68 were well-expressed across all three viruses at 72 h post reactivation (Fig 4C). Next, we tested whether the reduction in late protein levels in the ORF29.stop virus was due to a transcriptional defect. RT-qPCR measurements were taken late during infection at 48 h (Fig 4D) and 72 h post reactivation (Fig 4E) of viral RNA from the K8.1 (late) and ORF66, ORF6, and ORF68 (early) loci, as well as cellular GAPDH. At both time points, we observed a decrease in K8.1 mRNA levels in the ORF29.stop virus, mirroring the expression defect that was observed for K8.1 at the protein level. Surprisingly, levels of the representative early gene transcripts were also generally reduced in the ORF29.stop virus compared to WT. This finding was intriguing given that the loss of ORF29 did not appear to impair the levels of early gene-encoded proteins.

To help reconcile this finding with what was observed at the protein level, mRNA samples were analyzed at an earlier time point in infection. RT-qPCR measurements taken at 24 h post reactivation showed that transcript levels of the early genes ORF6 and ORF66 were not significantly different between WT and ORF29.stop virus (Fig 4F). These data indicate that the viral expression defects of the ORF29.stop virus occur between 24 and 48 h post reactivation. This period corresponds with the time window in which ORF29 protein appears, suggesting that the loss of ORF29 is linked to the general reduction in total viral mRNA levels observed.

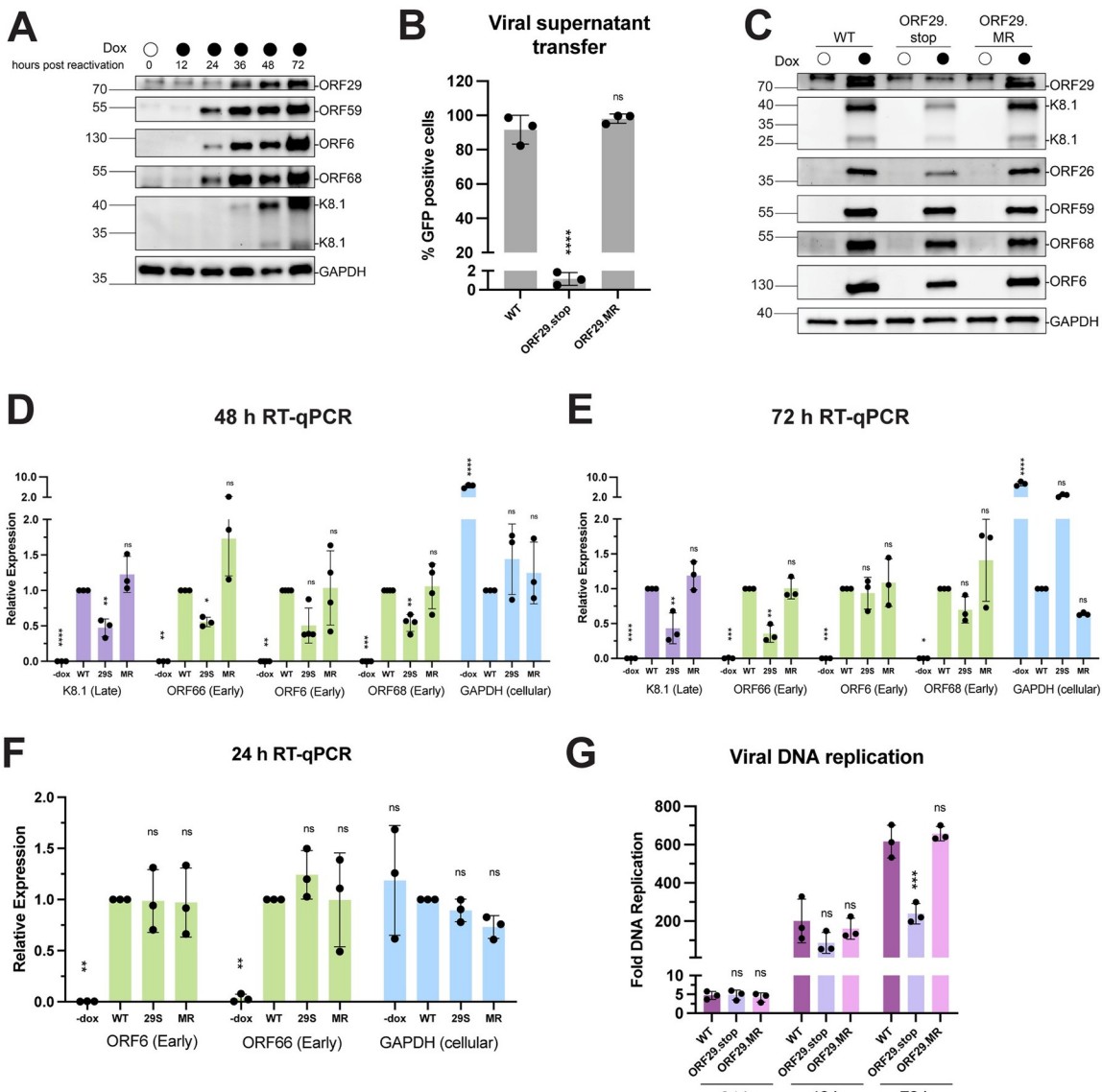

**Fig 4. Loss of ORF29 impairs late gene transcription.** (A) Western blot showing the expression kinetics of ORF29 relative to the early proteins ORF59, ORF6, and ORF68, and the late protein K8.1 in reactivated WT iSLK cells. (B) Infectious virion production from reactivated iSLK cells was measured by viral supernatant transfer using flow cytometry. Data are from three independent biological replicates, **** = P<0.0001. P values calculated from ordinary one-way ANOVA test. (C) Representative western blots of lysates harvested from iSLK cells at 72 h post reactivation. Levels of the late proteins, K8.1 and ORF26, are shown, as are levels of the early proteins, ORF6, ORF59, and ORF68. GAPDH serves as a loading control. (D) RNA levels of the indicated genes were measured by RTqPCR. Reactivated cells were harvested at 48 h post reactivation. Data are from three independent biological replicates, normalized to WT for each transcript, **** = P<0.0001, ** = P<0.005, * = P<0.05. P values calculated from ordinary one-way ANOVA test. (E) RNA levels of the indicated genes were measured by RTqPCR. Reactivated cells were harvested at 72 h post reactivation. Data are from three independent biological replicates, normalized to WT for each transcript, **** = P<0.0001, ** = P<0.005, * = P<0.05. P values calculated from ordinary one-way ANOVA test. (F) RNA levels of the indicated genes were measured by RTqPCR. Reactivated cells were harvested at 24 h post reactivation. Data are from three independent biological replicates, normalized to WT for each transcript, **** = P<0.0001, ** = P<0.005, * = P<0.05. P values calculated from ordinary one-way ANOVA test. (G) Viral DNA replication was measured using qPCR in cells harvested 24, 48, and 72 h post reactivation. Fold DNA replication values were calculated by comparing reactivated to non-reactivated cells. Data are from three independent biological replicates, * = P<0.05. P values calculated from ordinary one-way ANOVA test.

We also considered the possibility that the ORF29.stop virus could have a viral DNA replication defect that influenced transcript levels. To this end, we tested the impact of the loss of ORF29 on the replication of the viral genome at several points after reactivation (Fig 4G). DNA replication initiated at ~24 h post reactivation, with WT, ORF29.stop and ORF29.stop. MR viruses all having a 4-5-fold genome copy number increase compared to non-reactivated cells. Total levels of replicated DNA remained comparable between the viruses at 48 h post reactivation. We observed a modest DNA replication defect in the ORF29.stop virus compared to WT and ORF29.MR viruses at 72 h post reactivation, in agreement with what was observed in the ORF29 CRISPR-knockout cell line. At this time, the ORF29.stop virus had levels of replicated DNA that were roughly half that of the WT and ORF29.MR viruses, although comparison with the 48 h timepoint indicates that total genome copy number continues to increase for the ORF29.stop virus as the lytic cycle progresses. These data suggest that the mutant virus experiences attenuated but ongoing DNA replication throughout late infection. Thus, the loss of ORF29 may cause a mild reduction in viral DNA replication; however, this phenotype only manifests after a general reduction in total viral mRNA levels in the ORF29.stop virus is already apparent. In summary, in the absence of ORF29, there is a generalized reduction in viral mRNA late in infection, but a selective impairment of late protein accumulation.

## ORF29 catalytic activity is required to rescue the late gene expression defects of an ORF29 stop virus

ORF29 homologs in other herpesviruses have well-characterized roles in packaging and cleaving viral DNA into newly formed capsids [1]. The N-terminus of the protein has been proposed to function as a motor for the translocation of DNA into the capsid [35], while the C-terminus likely contains a nuclease domain that mediates cleavage of the DNA into unit length genomes [36,37].

To investigate whether ORF29's known roles as a DNA motor and nuclease during packaging were also involved in potentiating late gene expression, we tested the ability of a panel of ORF29 mutants to complement the ORF29.stop virus. ORF29 is the most highly conserved gene within the *Herpesviridae* family [38], and we leveraged this high degree of sequence conservation to generate predicted ORF29 functional mutants based on mutants that have been experimentally characterized in other viruses (S4A Fig). The N-terminal region of ORF29 contains a Walker A motif (ATP binding) and Walker B motif (ATP hydrolysis). Together, these sequences enable ATP metabolism that is thought to drive the motor function of the protein [35]. These sequences were mutated individually (Walker A: G226A, Walker B: E321K) and in tandem (G226A/E321K) to create putative motor-dead variants of ORF29. A recent study using purified ORF29 C-terminal nuclease domain concluded that the highly conserved residue D476 is required for nuclease activity *in vitro* [39]; therefore, we also complemented the ORF29.stop virus with D476A ORF29 (Fig 5A). ORF29.stop cells were additionally transduced with either WT ORF29 or empty vector as positive and negative controls, respectively.

WT ORF29 was able to complement the ORF29.stop virus, but none of the ORF29 mutants were able to complement this loss, as expected given the requirement for terminase function in virion formation (Fig 5B). To evaluate the role of these functional residues in late gene expression, we evaluated the expression of several representative early and late proteins by western blot in reactivated cells at 72 h post reactivation (Fig 5C). Notably, WT ORF29 but none of the ORF29 mutants consistently rescued K8.1 late protein expression in the ORF29. stop virus infections, while expression of early proteins ORF59 and ORF6 were not affected by loss or mutation of ORF29. Viral transcript levels were generally reduced across early and late genes for all mutants, matching the phenotype of the ORF29.stop virus (Fig 5D). Finally, we

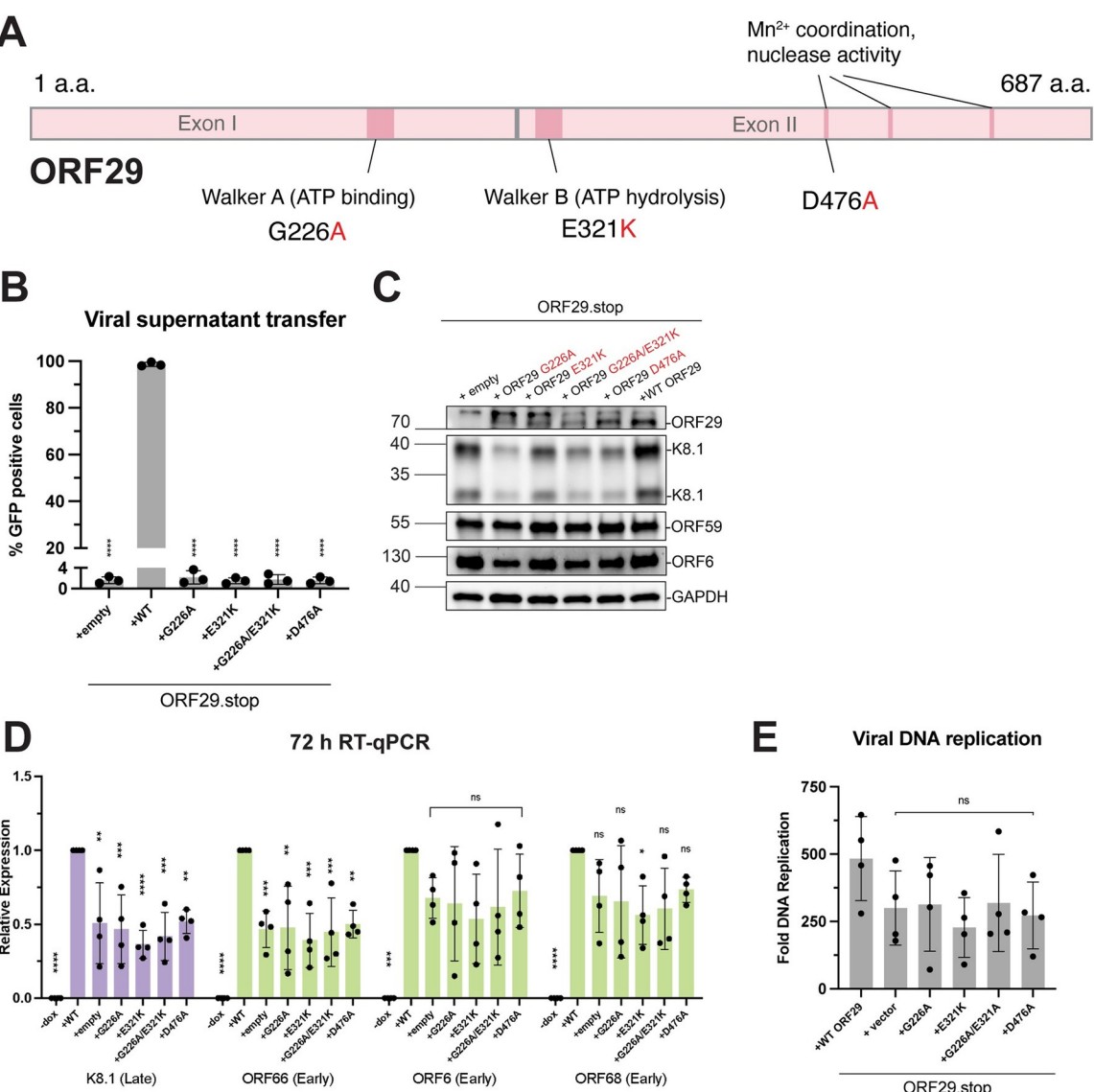

**Fig 5. Complementation with ORF29 functional mutants fails to rescue late gene transcription in the background of an ORF29. stop virus.** (A) Schematic of the ORF29 coding sequence. Positions of the Walker A and Walker B sequences are indicated, as are the residues that have previously been implicated in magnesium coordination and nuclease activity. The point mutants used in subsequent experiments are indicated in red. (B) Infectious virion production from reactivated iSLK cells was measured by viral supernatant transfer using flow cytometry. Data are from three independent biological replicates, **** = P<0.0001. P values calculated from ordinary one-way ANOVA test. (C) Representative western blots of lysates harvested from iSLK cells harboring the ORF29.stop virus complemented with ORF29 variants at 72 h post reactivation. Levels of the late protein K8.1 and early proteins ORF59 and ORF6 are shown. GAPDH serves as a loading control. (D) RNA levels of the indicated genes were measured by RTqPCR. Reactivated cells were harvested at 72 h post reactivation. Data are from three independent biological replicates, normalized to WT for each transcript, ** = P<0.005. P values calculated from ordinary one-way ANOVA test. (E) Viral DNA replication was measured using qPCR in cells harvested 72 h post reactivation. Fold DNA replication values were calculated by comparing reactivated to non-reactivated cells. Data are from three independent biological replicates. P values calculated from ordinary one-way ANOVA test.

observed no significant difference in total replicated DNA for the mutants compared to complemented WT ORF29 at 72 h post reactivation (Fig 5E). Collectively, our data reveal an unanticipated role for the catalytic subunit of the viral terminase in potentiating KSHV transcription and late protein expression, via a mechanism that requires its motor function and nuclease activity.

## Discussion

Mechanistic dissection of late gene transcription in β- and γ-herpesviruses has been limited by a lack of functional information on the vTAs and a paucity of additional factors known to contribute to this process. While viral DNA replication is a well-established requirement for late gene transcriptional activation, here we made the unexpected discovery that a core component of the herpesviral packaging machinery is also involved. Our proximity labeling and genetic approaches established that ORF29, the putative catalytic component of the KSHV terminase, is both associated with the vTA complex and contributes functionally to gene expression late in infection. These insights indicate that ORF29 plays a previously unappreciated role in viral gene expression and suggests that the essential processes of transcription, genomic replication, and packaging occurring late during infection are more intimately interconnected than previously thought.

ORF29 is part of the tripartite terminase complex that threads replicated viral DNA into nascent capsids and cleaves the concatenated genome into unit-length genomes. In KSHV, the other terminase subunits are ORF7 and ORF67.5. ORF7 contributes to DNA cleavage and capsid formation [30], while the function of ORF67.5 is unknown, though its homolog in HSV-1 has been proposed to function in terminase holoenzyme assembly [40]. Neither ORF7 nor ORF67.5 were enriched in the proximity labeling screen. This could be due to the limitations of the assay, or because ORF29 potentiates late gene expression independently of its function within the terminase complex. Two observations favor the latter hypothesis. First, an ORF7-null virus did not demonstrate viral expression defects at the mRNA or protein level across all kinetic classes [30]. Second, we previously showed that loss of the essential packaging protein ORF68 also did not affect viral gene expression [41]. These data suggest that there is an ORF29-specific contribution to viral gene expression that is lost in the ORF29.stop mutant, rather than the defect stemming from a disruption to the terminase complex or other facets of the packaging process. However, future work examining the impact of the loss of functional packaging machinery is necessary to explore this possibility more completely.

Mutation or deletion of ORF29 promotes a generalized decrease in early and late viral mRNA abundance, but a selective decrease in late protein levels. We hypothesize that this stems from the fact that early mRNAs are produced and translated abundantly prior to KSHV DNA replication (and continue to be transcribed later), whereas late mRNAs are only transcribed at the end stages of infection. Thus, impairing transcription after the onset of viral DNA replication should have a significantly more pronounced effect on the accumulation of proteins encoded by late genes. Differences in protein and RNA half-lives and the timing of ORF29 induction may also contribute to this phenotype. Alternatively, ORF29 may promote the selective accumulation of late proteins though a yet uncharacterized mechanism.

How might ORF29 facilitate transcription at late stages of infection? ORF29 mutants putatively defective in motor and nuclease function fail to rescue the defects in late protein expression that occur in the ORF29.stop virus, suggesting that its catalytic activity is involved. As a factor that can both move along and cleave DNA, ORF29 may be processing the newly replicated viral DNA to keep the genome in a state that is permissive for transcription (Fig 6A). A second possibility arises from work showing that DNA torsional strain can impair transcription [42–44]. Torsional stress is generated by many DNA-based processes including genome replication and transcription and is relieved by topoisomerases that act by cleaving DNA. Intriguingly, topoisomerase inhibition has been shown to affect aspects of RNA polymerase II kinetics and impair transcription [42,43]. Given that ORF29 regularly cleaves DNA during packaging, it is tempting to speculate that a similar relationship could be at play (Fig 6B).

Perhaps as illuminating as the identification of ORF29 from the vTA complex proximity labeling screen is the set of factors that were not selectively enriched under these conditions.

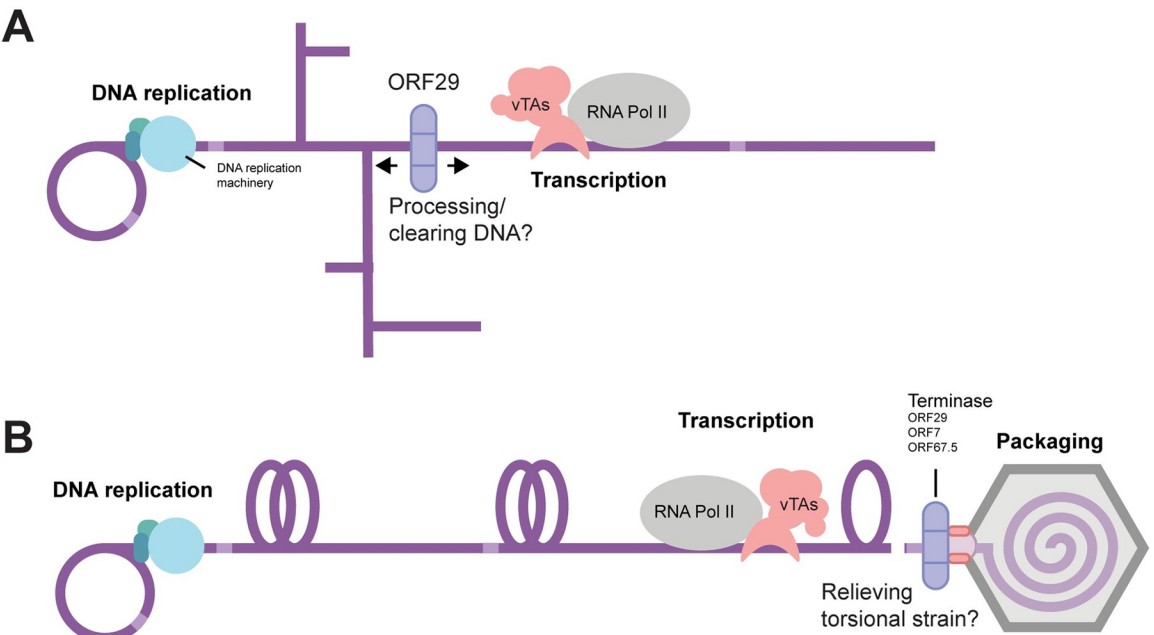

**Fig 6. Models for the contribution of ORF29 to viral transcription.** (A) Newly replicated concatemeric herpesvirus genomes are shown with aberrant DNA structures that are thought to arise during replication. As a factor that can both move along and cleave DNA, ORF29 may function to resolve DNA structures that could stall RNA polymerase or are otherwise detrimental to transcription. Light purple regions on the DNA denote terminal repeats at the end of each unit length genome. (B) DNA replication and transcription can generate torsional strain shown as supercoiled DNA. If left unresolved, torsional strain can impair transcription. ORF29 activity may relieve accumulated torsional strain to allow for full viral transcription.

While there are many reasons why proteins may have escaped detection or fallen below the enrichment threshold, the total absence of cellular general transcription factors (GTFs) is conspicuous. The cellular PIC can contain upwards of forty individual proteins [45]; that none aside from subunits of RNA polymerase II were enriched near the vTAs is striking. While this finding does not rule out the possibility that other cellular factors may be involved in late gene transcription, it adds to a growing body of evidence suggesting these proteins do not occupy the promoter at the same time as the vTAs [12,46].

A previously published study in our lab systematically mapped the protein-protein interaction network between KSHV proteins and the human proteome and identified ORF24 as binding RNA polymerase II but not to any other cellular PIC components [12]. However, chromatin immunoprecipitation experiments from the same study revealed that the general transcription factors TFIIB and TFIIH, which respectively modulate promoter binding and opening, are present at late gene promoters. A recent study in human cytomegalovirus (HCMV) characterized PICs driven by TBP versus the HCMV homolog of ORF24, UL87, on the viral genome [46]. These transcriptional complexes had distinct sizes and divergent positioning around transcription start sites (TSS). TBP-PICs were larger and straddled the TSS, corresponding to the footprint of TFIID, a multi-subunit complex which contains TBP and mediates polymerase recruitment and promoter recognition. In contrast, UL87 PICs were smaller and upstream of the TSS, suggesting that they lack the subunits of TFIID and TFIIH that contact downstream DNA. However, the same study showed that transcription initiation from UL87 PICs is sensitive to TFIIH inhibition. Together, these data and our work support a model in which the vTAs and a subset of general transcription factors occupy late gene promoters but do so without closely associating with each other. One possibility is that the vTA

complex binds to late gene promoters, either precomplexed with RNA polymerase II or recruiting the polymerase after binding, then dissociates prior to subsequent binding of other general transcription factors. Future work should seek to determine the definitive set of general transcription factors that are recruited to late gene promoters to gain a better understanding of how this process is orchestrated.

DNA packaging in herpesviruses is thought to be mechanistically similar to that of tailed bacteriophages [47]. ORF29, like all its herpesvirus homologs, shares sequence elements and structural features with the large catalytic subunit of the bacteriophage T4 terminase, gp17 [48]. Interestingly, work in the T4 bacteriophage system has shown that the gp17 terminase directly interacts with the phage sigma factor, gp55, which is not only part of the phage late transcription apparatus but also engages the T4 sliding clamp [49,50]. These interactions hint at potential mechanistic coupling between the late processes of DNA replication, late gene transcription, and packaging in T4 bacteriophage. In light of our findings that ORF29 contributes to both gene expression and packaging late during infection, and the parallels between bacteriophage and herpesvirus terminases, we consider the possibility that ORF29 lies at the intersection of at least two essential viral processes in KSHV. In this regard, future inhibitors developed against ORF29 have the potential to restrict multiple stages of the γ-herpesviral lifecycle.

## Materials and methods

### Plasmids

All plasmids generated in this study have been deposited on Addgene. ORF18, ORF30, ORF31, and ORF66-TurboID were subcloned into the BamHI and NotI sites of pcDNA4 3xHA (C-term) using InFusion cloning (Clontech) (Addgene 200017–200020). TurboID-ORF34 was subcloned into the NotI and XhoI sites of N-term 3xHA pcDNA4 (Addgene 200021). Plasmids pcDNA4/TO-ORF18-2xStrep (Addgene 120372), pcDNA4/TO-ORF24-2xStrep (Addgene 129742), pcDNA4/TO-ORF30-2xStrep (Addgene 129743), pcDNA4/TO-ORF31-2xStrep (Addgene 129744), pcDNA4/TO-ORF66-2xStrep (Addgene 130953) and pcDNA4/TO-2xStrep-ORF34 (Addgene 120376) have been previously described [7,8]. Plasmid K8.1 Pr pGL4.16 + Ori (Addgene 131038) and ORF57 Pr pGL4.16 (Addgene 120378) have been previously described [7]. Plasmid pRL-TK (Promega) was kindly provided by Russell Vance. 3xHA-TurboID-NLS was subcloned from 3xHA-TurboID-NLS_pCDNA3 (Addgene 107171) into the AgeI/BamHI sites of pLVX-TetOne-zeo using InFusion cloning to generate pLVX-TetOne-zeo 3xHA TurboID-NLS (200022). The bleomycin resistance marker and T2A-mCherry were subcloned into the NheI/EcoRI sites pMCB320 sgRNA lentiviral delivery plasmid (Addgene 89359) using InFusion cloning to generate pDM002-sgRNA-lenti-ZeoR (Addgene 200060). Guide sequences targeting cellular genes were generated using CRIS-Pick (Broad Institute). Guide sequences targeting viral ORFs and the safe-targeting viral guides were generated using the Integrated DNA Technologies (Integrated DNA Technologies) custom guide RNA tool. Safe-targeting cellular guides and non-targeting guide sequences have been previously published [51]. Oligos for each guide sequence (Integrated DNA Technologies) were ligated using T4 ligase into the BstXI/BlpI sites of pDM002-sgRNA-lenti-ZeoR, a mU6-driven guide expression plasmid that had been subcloned to confer zeocin resistance. This plasmid also constitutively expresses mCherry. All sgRNA sequences used in this study are listed in S2 Table. Three separate guides were cloned per target gene. KSHV ORF29 was subcloned into the AgeI and EcoRI sites of pLJM1 that had been modified to confer zeocin resistance (Addgene 200028). Point mutations in pLJM1-zeo ORF29 were generated using inverse PCR site-directed mutagenesis with Phusion DNA polymerase (New England Biolabs)

(Addgene 200024–200027). Primers for these reactions are listed in S3 Table. Plasmids lenti-Cas9-Blast (Addgene 52962), pMD2.G (Addgene 12259), pMDLg/pRRE (Addgene 12251), pRSV-Rev (Addgene 12253), and psPAX2 (Addgene 12260) have been described previously. 3xHA-TurboID-NLS_pCDNA3 (Addgene 107171) was a gift from Alice Ting. pMCB320 (Addgene 89359) was a gift from Michael Bassik. lentiCas9-Blast was a gift from Feng Zhang. Lentiviral packaging plasmids were a gift from Didier Trono.

## KSHV BAC16 mutagenesis

All recombinant KSHV mutants were engineered using the scarless Red recombination system in BAC16 GS1783 *E. coli* as previously described [24]. The modified BACs were purified using a Nucleobond BAC 100 kit (Clonetech), and successful incorporation of the mutations was confirmed by colony PCR. BAC integrity was assessed by digestion with the restriction enzymes RsrII and SbfI (New England Biolabs) and by full plasmid sequencing (Plasmidsaurus).

## Cell line maintenance

All HEK293T cells (ATCC) were minimally maintained in Dulbecco modified Eagle medium (DMEM) supplemented with 10% fetal bovine serum (FBS) (Seradigm). HEK293T cells constitutively expressing ORF29 were additionally maintained in 500 µg/ml zeocin. iSLK-puro cells were maintained in DMEM supplemented with 10% FBS and 1 µg/ml puromycin. iSLK-puro cells and the iSLK cell line harboring the KSHV genome on the bacterial artificial plasmid BAC16 and a doxycycline-inducible copy of the lytic transactivator RTA (iSLK BAC16 cells) have been previously described [24]. All iSLK cell lines were minimally maintained in DMEM supplemented with 10% FBS, 1µg/ml puromycin, and 1 mg/ml hygromycin. TurboID-NLS BAC16 iSLKs, and all ORF29.stop BAC16 complemented iSLK lines were additionally maintained in 500 µg/ml zeocin. iSLKs transduced with Cas9 and sgRNA expression plasmids were additionally maintained in 10 µg/ml blasticidin and 500 µg/ml zeocin.

## Generation of iSLK cell lines

Generation of iSLK cell lines latently infected with modified KSHV virus was accomplished by transfecting HEK293T cells (WT HEK293T for the TurboID-ORF18 BAC, early and late reporter BACs; HEK293T transduced to stably express ORF29 for the ORF29.stop BAC) with 5 µg recombinant BAC DNA using PolyJet (SignaGen). HEK293T cells stably expressing ORF29 were generated via lentiviral transduction to propagate the ORF29.stop virus, which had lost a gene essential for replication. One day post transfection, the transfected HEK293T cells were trypsinized and cocultured 1:1 with trypsinized iSLK-puro cells. Cocultures were treated with 30 nM 12-*O*-tetradecanoylphorbyl-13-acetate (TPA) and 300 mM sodium butyrate for 4 days to induce lytic replication and allow for infection of the SLK cells with KSHV. Successfully infected cells were selected in media containing 1 µg/ml puromycin, 250 µg/ml G418, and 300 µg/ml hygromycin. The hygromycin concentration was incrementally increased to 500 µg/ml, and 1 mg/ml until all 293T cells had died.

## Transduction of iSLK cell lines

ORF29.stop iSLKs were transduced with empty pLJM1 vector, WT ORF29, or ORF29 mutants. WT iSLKs were transduced with pLVX-TetOne-zeo TurboID-NLS. All lentivirus was generated in HEK293Ts cotransfected with the aforementioned plasmids along with packaging plasmids pMD2.G and psPAX2 using PolyJet reagent (SignaGen). The supernatant was harvested

48 h post-transfection and syringe filtered through a 0.45 μm pore-sized filter (Millipore). The filtered supernatant was diluted 1:2 in serum-free DMEM. Polybrene (Millipore) was added to a final concentration of 8 μg/ml, and the supernatant was added to freshly trypsinized target cells in 6 well plates. Cells were spinoculated for 2 h at 1000 x $g$ at 37˚C. Following transduction, cell lines were selected in their respective maintenance media supplanted with 500 μg/ml zeocin (Sigma).

## Preparation of cell lysates containing biotinylated proteins

WT iSLKs, ORF18-TurboID-3xHA iSLKs, and TurboID-NLS iSLKs grown on 15 cm plates were reactivated with 1 μg/ml doxycycline and 1 mM sodium butyrate. 48 hours later, cells were washed with PBS and media was changed to 10% FBS DMEM supplemented with 500 μM biotin (Sigma Aldrich) in DMSO, or DMSO alone. Plates were incubated for 10 minutes at room temperature before the media was aspirated away and replaced with fresh DMEM with 10% FBS, and the plates were allowed to sit for an additional 30 minutes at room temperature. Cells were washed and collected in ice cold PBS and centrifuged at 300 x $g$ for 5 min at 4˚C. Cell pellets were resuspended in 1 mL RIPA lysis buffer (50mM Tris-HCl pH 8, 150 mM NaCl, 0.1% SDS, 0.5% sodium deoxycholate, 1% Triton X-100) with cOmplete EDTA-free Protease Inhibitor cocktail (Roche) and lysed by rotating for 1 h at 4˚C. Samples were treated with Benzonase nuclease (Millipore-Sigma) followed by sonication using a QSonica Ultrasonicator with a cup horn set to 100 amps for 1 min total (3 seconds on, 17 seconds off). Lysates were then clarified by centrifugation at 21,000 x $g$ for 10 min at 4˚C and quantified by Bradford assay.

## Affinity purification of biotinylated proteins

50 μl of Pierce Streptavidin magnetic beads (Thermo Fisher) for each sample were prewashed with RIPA lysis buffer. A total of 4 mg whole cell lysate was incubated with prewashed beads and rotated for 1 h at room temperature followed by rotation overnight at 4˚C. The beads were collected using a magnetic holder and washed with 1 mL of buffer as follows at room temperature: RIPA lysis buffer (twice, 2 min per wash), 1M KCl (once, 2 min), 0.1M sodium carbonate (once, 2 min), 2 M urea in 10 mM Tris-HCl pH 8.0 (once, 10 sec), RIPA lysis buffer (twice, 2 min per wash). Samples for mass spectrometry (MS) were resuspended in a final volume of 1 mL RIPA lysis buffer and sent to the Cristea Lab at Princeton University for analysis. Samples for western blot analysis were eluted with 4X dye-free sample buffer (250 mM Tris–HCl (pH 6.8), 8% (w/v) SDS, 40% (v/v) glycerol, 20% (v/v) β-mercaptoethanol) and 10mM biotin.

## TurboID sample preparation for MS

Samples were stored at 4˚C until ready to proceed with the streptavidin IP. Beads were transferred to a new LoBind tube (Amuza, Inc. (Eicom USA)), and the RIPA buffer was aspirated. Beads were then washed twice with 200 μL 50 mM Tris HCl pH 7.5 and twice with 1 mL 2 M urea in 50 mM Tris pH 7.5. An on-bead digestion of the 4 mg sample was performed by resuspending the beads in 80 μL 2 M urea in 50 mM Tris pH 7.5 with 1 mM DTT and 3.2 μg trypsin and incubating for 1 h at 25˚C while shaking at 600 rpm. Supernatant was transferred to a new LoBind tube, the beads were washed twice with 60 μl 2 M urea in 50 mM Tris pH 7.5, and the washes were combined with the on-bead digest supernatant. Samples were reduced and alkylated with 25 mM TCEP (Tris(2-carboxyethyl)phosphine) (Thermo Fisher) and 50 mM CAM (chloroacetamide) (Fisher Scientific) by shaking at 600 rpm in a thermomixer at 25˚C for 1.5 h. Samples were diluted 1:1 (v/v) with 50 mM Tris pH 8 and an additional 4 μg trypsin were added. Samples were digested overnight at 25˚C with shaking at 600 rpm. Samples were acidified to 1% TFA, incubated on ice for 15 min, and spun down at 4000 x$g$ for 5 min at 4˚C.

Samples were then desalted over C18 StageTips (Empore™ SPE Disks C18, Fisher Scientific). Samples were dried in a speedvac and resuspended in 6 μL 1% formic acid/1% acetonitrile.

## MS data acquisition

IP samples were run on a Q-Exactive HF mass spectrometer (ThermoFisher Scientific) equipped with a Nanospray Flex Ion Source (ThermoFisher Scientific). Peptides were separated on an in-house 50 cm column (360 μm od, 75 μm id, Fisher Scientific) packed with ReproSil-Pur C18 (120 Å pore size, 1.9 μm particle size, ESI Source Solutions). Peptides were separated over a 150 min gradient of 5% B to 30% B at a flow rate of 0.250 nl/min. MS1 scans were collected with the following parameters: 120,00 resolution, 30 ms MIT, 3e6 AGC, scan range 350 to 1800 m/z, and data collected in profile. MS2 scans were collected with the following parameters: 30,000 resolution, 150 ms MIT, 1e5 AGC, 1.2 m/z isolation window, loop count of 15, NCE of 28, 100.0 m/z fixed first mass, peptide match set to preferred, and data collected in profile at a dynamic exclusion of 30 s.

## MS data analysis

MS/MS spectra were analyzed in Proteome Discoverer v2.4 (Thermo Fisher Scientific). Sequest HT was used to search spectra against a Uniprot database containing human (downloaded January 2021) and KSHV protein sequences (downloaded March 2021) and common contaminants. Offline mass recalibration was performed via the Spectrum Files RC node, and the Minora Feature Detector node was used for label-free MS1 quantification. Fully tryptic peptides with a maximum of two missed cleavages, a 4 ppm precursor mass tolerance, and a 0.02 Da fragment mass tolerance were used in the search. Posttranslational modifications (PTMs) that were allowed included the static modification carbamidomethylation of cysteine, and the dynamic modifications of oxidation of methionine, deamidation of asparagine, loss of methionine plus acetylation of the N-terminus of the protein, acetylation of the N-terminus of the protein, acetylation of lysine, and phosphorylation of serine, threonine, and tyrosine. Peptide spectrum match (PSM) validation was done using the Percolator node and PTM sites were assigned in the ptmRS node. PSMs were assembled into peptide and protein identifications with a false discovery rate of less than 1% at both the peptide and protein level with at least 2 unique peptides identified per protein. Samples were normalized using the total peptide amount, protein abundances were calculated using summed abundances, and protein ratio calculations were performed using pairwise ratios.

Proteins were further filtered using Microsoft Excel. For a protein to be considered a putative interactor of ORF18, the protein had to have at least a twofold change in abundance when compared to all three control samples (WT—ligase, TurboID-NLS, and ORF18-TurboID—biotin) and had to have at least two peptides quantified in the ORF18-TurboID + biotin sample. The final interaction dataset contains the proteins that passed all filtering steps in all three replicates. Subcellular localization was assigned based on Uniprot annotations. Protein networks were generated using STRING (v.11) [52] and Cytoscape (v.3.8.2) [53]. Gene ontology term analysis was carried out using the PANTHER Classification System.

## CRISPR screen

Cas9 was stably introduced to the early and late gene reporter cell lines by lentiviral transduction. lentiCas9-blast was transfected with 3rd generation lentiviral packaging plasmids into HEK293T cells using PolyJet reagent (SignaGen). The supernatant was harvested 48 h posttransfection and syringe filtered through a 0.45 μm pore-sized filter (Millipore). The filtered supernatant was diluted 1:2 in serum-free DMEM. Polybrene (Millipore) was added to a final

concentration of 8 μg/ml, and the supernatant was added to $1\times10^6$ iSLK reporter cells. The reporter cells were inoculated by spinning in a 6-well plate at 500 x g for 2 h at 37°C. 48 h post-inoculation, the transduced reporter cells were expanded to 10 cm plates and maintained in media containing 1 mg/mL hygromycin, 1 μg/mL puromycin, and 10 μg/ml blasticidin for 2 weeks to select for Cas9 integration.

For each protein-coding target identified in the MS screen, three separate guides were generated. Lentivirus was produced from HEK293Ts for each sgRNA plasmid, then supernatants were pooled for guides with the same genetic target before filtration and application to Cas9 + reporter iSLK cells. 48 h post-transduction, reporter cells were placed under zeocin selection (500 μg/mL) and additionally maintained in 10 μg/ml blasticidin, 1 mg/ml hygromycin and 1 μg/ml puromycin. Cells were expanded 7 days post-transduction.

To carry out the fluorescent reporter assays, knockout cells for each target were plated and reactivated with 5 μg/mL doxycycline and 1 mM sodium butyrate. Cells for the late gene reporter assay were harvested at 72 h post-reactivation, and cells for the early gene reporter assay were harvested at 48 h post-reactivation. Media was aspirated from cells that were then washed in PBS, trypsinized, and fixed in 4% paraformaldehyde diluted in PBS. After crosslinking, cells were pelleted and resuspended in PBS. GFP, mIFP, and mCherry signal was quantified by flow cytometry on a BD Bioscience LSR Fortessa. 10,000 events were collected per sample, and replicates represent independent reactivations on separate days. All data were analyzed using FlowJo software (BD Bioscience).

## Supernatant transfer assay

Supernatant from reactivated iSLKs was syringe filtered through 0.45 μm filters 72 h post reactivation and 2 mL of supernatant was supplied to naive HEK293T cells. HEK293Ts were inoculated for 2 h under constant spinning at 1000 x *g*. 24 h post inoculation, the media was aspirated and the HEK293Ts were washed with cold PBS and fixed in 4% paraformaldehyde diluted in PBS. After crosslinking, cells were pelleted and resuspended in PBS. GFP signal was quantified by flow cytometry on a BD Bioscience LSR Fortessa. 10,000 events were collected per sample, and replicates represent independent reactivations on separate days. All analysis was performed using FlowJo software.

## Western blotting

Cells were washed and pelleted in cold PBS, followed by resuspension in lysis buffer (150 mM NaCl, 50 mM Tris-HCl, pH 7.4, 1mM EDTA, 0.5% NP-40, and protease inhibitor (Roche)). Cell lysates rotated for 1 h at 4°C and were clarified by centrifugation at 21,000 x *g* for 10 min. 10–20 μg whole cell lysate were used for SDS-PAGE. Western blotting was done in Tris-buffered saline and 0.1% Tween 20 (TBS-T) using the following primary antibodies: rabbit anti-ORF59 (1:10,000), rabbit anti-K8.1 (1:10,000), rabbit anti-vinculin (1:1,000; Abcam), streptavidin-HRP (1:500 Thermo Fisher), rabbit anti-XRN1 (1:1,000; Bethyl), mouse anti-CASK (1:1,000; Abcam), rabbit anti-PABPC1 (1:1,000; Cell Signaling Technologies), rabbit anti-ORF6 (1:10,000), rabbit anti-PABPC4 (1:1,000; Bethyl), mouse anti-ORF26(1:500; Novus), rabbit anti-ORF68 (1:5,000), mouse anti-GAPDH (1:1,000; Abcam), rabbit anti-ORF29 (1:500). Rabbit anti-ORF59, K8.1, ORF6, and ORF68 have all been previously described [8, 54]. Preparation of the rabbit anti-ORF29 antibody is described below.

## Quantification of viral DNA replication

Fold DNA induction was determined using reactivated and non-reactivated iSLK cells that were rinsed with PBS, scraped, pelleted at 300 x g for 5 min at 4°C, and resuspended in 200 μl

PBS. Genomic DNA was extracted using a NucleoSpin Blood kit (Macherey-Nagel) following the manufacturer's protocol. Fold viral DNA replication was quantified by qPCR using iTaq Universal SYBR green Supermix (Bio-Rad) on a QuantStudio3 Real-Time PCR machine. Primers specific to the KSHV ORF57 promoter were used to quantify viral DNA and normalized to the human GAPDH promoter and to non-reactivated samples to determine fold replication.

## RT-qPCR

Total RNA was extracted in TRIzol (Invitrogen) and isolated using either a Direct-Zol RNA miniprep plus kit (Zymo) or chloroform extraction. Purified RNA was treated with Turbo DNase (Thermo Fisher) and reverse-transcribed using AMV RT (Promega). cDNA was quantified by qPCR using iTaq Universal SYBR green Supermix (Bio-Rad) on a QuantStudio3 Real-Time PCR machine using transcript-specific primers. Transcript levels were normalized to the 18S rRNA signal.

## siRNA transfections

Transfection of siRNAs into WT BAC16 iSLKs was performed with Lipofectamine RNAiMax reagent (Invitrogen) following the manufacturer's protocol. $0.5 \times 10^6$ cells were seeded and transfected the following day with a pool of 4 siRNAs or a non-targeting control using either 5 or 10 μl of 20 μM siRNA (Dharmacon). Cells were reactivated 24 h post-transduction and harvested 48 h post-reactivation for lysis and western blotting.

## Late gene reporter assay

$1 \times 10^6$ HEK293T cells were plated in 6 well plates. After 24 h, each well was transfected with 900 ng total DNA containing 125 ng each of either pcDNA4-2xStrep wild-type or pcDNA4 3xHA-TurboID-tagged ORF18, ORF30, ORF31, ORF34, ORF66, and pcDNA4/TO-ORF24-2xStrep, or 750 ng of empty pcDNA4/TO-2xStrep as a control, along with either K8.1 Pr pGL4.16 or ORF57 Pr pGL4.16 and 25 ng of pRL-TK as a transfection control. 24 h post-transfection, cells were rinsed twice with PBS, lysed by rocking for 15 min at room temperature in 500 μl of passive lysis buffer (Promega), and clarified by centrifugation for 2 min at 21,000 x *g*. 20 μl of lysate was plated in triplicate to a white chimney well microplate (Greiner Bio-One), and luminescence was measured on a Tecan M1000 microplate reader using a dual-luciferase assay kit (Promega). Firefly luminescence was normalized to the renilla luminescence to control for each transfection. Results for all samples were normalized to those for the corresponding control containing empty plasmid.

## ORF29 purification and antibody generation

The nuclease domain of ORF29 (residues 442–687) containing an N-terminal 6xHis tag and TEV site was produced in *Escherichia coli* NiCo21 (DE3) cells (New England Biolabs) as previously described in https://www.ncbi.nlm.nih.gov/pmc/articles/PMC6069193/. Purified ORF29b was used for immunization of two rabbits by YenZym Antibodies LLC (Brisbane, CA). Sera was purified in-house over a HiTrap Protein G column (Cytiva Life Sciences) followed by positive and negative selection with ORF29b-AminoLink and MBP-AminoLink columns generated using the AminoLink Plus Coupling Resin kit (Thermo Scientific) according to the manufacturer's instructions. The final ORF29b antibody was concentrated to 1 mg/mL and used at a final concentration of 1 μg/mL.

## Supporting information

**S1 Table. MS dataset from Turbo-ID experiment.** All cellular and viral proteins identified across all IPs and replicates are shown. Tab 1 outlines the information displayed in each column.
(XLSX)

**S2 Table. List of CRISPR guide sequences used in this study.**
(XLSX)

**S3 Table. List of all DNA sequences used in this study.**
(XLSX)

**S1 Fig. Construction of the ORF18-TurboID BAC16 virus.** (A) Western blots of HEK293T cells transiently transfected with vTA-TurboID-3xHA constructs indicating that all vTAs tested express when fused to the ligase. (B) HEK293T cells were transiently transfected with plasmids encoding the six vTAs (either WT or TurboID-tagged as indicated), a plasmid containing the firefly luciferase gene under control of either the ORF57 or K8.1 promoter, and a plasmid encoding renilla luciferase as a transfection control. (C) Diagram showing the genomic locus of ORF18 and the site of introduction of TurboID-3xHA. ORF18 partially overlaps with ORF17 and is directly upstream of ORF19. (D) The recombinant ORF18-TurboID BAC was digested by the restriction enzymes SbfI and RsrII and compared to digested WT BAC16, which confirmed that the introduction of TurboID had not resulted in any large-scale recombination events within the BAC. (E) Western blots of iSLK lysates indicating that the addition of TurboID to ORF18 within the BAC does not impair expression of representative early or late proteins upon reactivation.
(TIF)

**S2 Fig. Structure of the early gene reporter virus.** (A) Schematic of the early gene reporter virus used to measure early gene expression by flow cytometry. The reporter construct was cloned into a region of the vector backbone of the KSHV BAC16. The ORF57 early gene promoter drives expression of eGFP, and mIFP is constitutively expressed.
(TIF)

**S3 Fig. Construction of the ORF29.stop and ORF29.stop MR viruses.** (A) Diagram showing the genomic locus of ORF29 and the site of introduction of the premature stop codons. ORF29 is composed of two exons; two premature stop codons were introduced near the start of the second exon. (B) The recombinant ORF29.stop and ORF29.MR BACs were digested by the restriction enzymes SbfI and RsrII and compared to digested WT BAC16, which confirmed that the introduction of these mutations had not resulted in any large-scale recombination events within the BAC.
(TIF)

**S4 Fig. Multiple sequence alignment of ORF29 homologs.** (A) Multiple sequence alignment of ORF29 and homologs from representative alpha-, beta-, and gammaherpesviruses. (**\***) indicates a fully conserved residue, (**:**) indicates conservation of residues with similar properties, (**.**) indicates conservation of residues with weakly similar properties. The Walker A and Walker B motifs are highlighted in green and blue respectively. The aspartic acid required for nuclease activity at position 476 in ORF29 is highlighted in purple. Alignment generated using the T-coffee method [55].
(TIF)

## Acknowledgments

We would like to thank David Morgens for his support in conceptualizing and carrying out the CRISPR work in this study, as well as all members of the Glaunsinger lab for helpful discussions and suggestions.

## Author Contributions

**Conceptualization:** Chloe O. McCollum, Allison L. Didychuk, Britt A. Glaunsinger.

**Data curation:** Dawei Liu, Laura A. Murray-Nerger.

**Formal analysis:** Chloe O. McCollum, Dawei Liu, Laura A. Murray-Nerger.

**Funding acquisition:** Ileana M. Cristea, Britt A. Glaunsinger.

**Investigation:** Chloe O. McCollum, Allison L. Didychuk, Dawei Liu, Laura A. Murray-Nerger.

**Methodology:** Chloe O. McCollum, Allison L. Didychuk, Dawei Liu, Laura A. Murray-Nerger.

**Project administration:** Britt A. Glaunsinger.

**Resources:** Ileana M. Cristea.

**Supervision:** Ileana M. Cristea, Britt A. Glaunsinger.

**Validation:** Chloe O. McCollum.

**Writing – original draft:** Chloe O. McCollum.

**Writing – review & editing:** Chloe O. McCollum, Allison L. Didychuk, Dawei Liu, Laura A. Murray-Nerger, Ileana M. Cristea, Britt A. Glaunsinger.

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
