## [Decision Letter · Decision Letter 0]

14 Mar 2023

Dear Dr. Glaunsinger,

Thank you very much for submitting your manuscript "The viral packaging motor potentiates late gene expression in Kaposi's sarcoma-associated herpesvirus" for consideration at PLOS Pathogens. As with all papers reviewed by the journal, your manuscript was reviewed by members of the editorial board and by several independent reviewers. The reviewers appreciated the attention to an important topic. All three reviewers were enthusiastic and found the manuscript to be innovative, significant and well performed technically. The recommendations for revision can mostly be addressed by textual revisions.  One reviewer's suggestion that an additional western blot be performed is not essential but would strengthen the paper.  Based on the reviews, we are likely to accept this manuscript for publication, providing that you modify the manuscript according to the review recommendations.

Sincerely,

Sankar Swaminathan, MD

Academic Editor

PLOS Pathogens

Patrick Hearing

Section Editor

PLOS Pathogens

Kasturi Haldar

Editor-in-Chief

PLOS Pathogens

orcid.org/0000-0001-5065-158X

Michael Malim

Editor-in-Chief

PLOS Pathogens

orcid.org/0000-0002-7699-2064

Reviewer Comments (if any, and for reference):

Reviewer's Responses to Questions

**Part I - Summary**

Reviewer #1: This manuscript by McCollum et al. uses the TurboID proximity ligation system to identify proteins associated with KSHV ORF18, one of the viral transcriptional activators (vTAs) required for late gene expression. The experimental design is rigorous. A recombinant KSHV with an ORF18-TurboID fusion is generated in BAC16 and tested to ensure early and late gene transcription are intact. In iSLK cells infected with this recombinant KSHV, a total of 45 proteins were found to be associated with ORF18 (but not in 3 appropriate control conditions). Included among these were all known "ORF18-interacting proteins." Using CRISPR knockouts they demonstrate that some host genes (CASK, PABPC1) selectively impaired late gene expression; however, siRNA to PABPC1 did not have this effect unless PABPC4 was also knocked down.

Next, they turn their attention to ORF29 - a surprising hit as this encodes the ATPase subunit of the viral terminase which was not previously suspected to play a role in late gene transcription. They find that an ORF29stop virus has decreased levels of late but not early proteins relative to WT. Surprisingly, they find that at the RNA level both early and late mRNAs are decreased at 48h and 72h; however, this reduction in early mRNAs relative to WT is not present a 24h post induction. ORF29 point mutations are tested by transcomplementation to demonstrate that sites implicated in ATP binding, ATP hydrolysis, and nuclease activity are all essential for restoring the early and late transcription defects in the ORF29stop background.

In summary, this manuscript describes a rigorous set of experiments that defines several viral and cell proteins that are in proximity to the KSHV ORF18 vTA. Follow-up experiments define CASK and PABPC1/PABPC4 as cell proteins important for late gene expression - a novel result. The finding that the ORF29 plays a role in maintaining both early and late transcription is interesting and suggests an unexpected integration between packaging and transcription. With some minor exceptions (see below), I think that their conclusions are supported by this data and would be of interest to PLoS Pathogens readers.

Reviewer #2: In this manuscript, McCollum and colleagues presented compelling results revealing the role of the viral protein ORF29 on viral transcription and late gene expression in Kaposi's sarcoma-associated herpesvirus, KSHV. ORF29 is the catalytic subunit of the KSHV DNA packaging complex. Using proximity labeling coupled with mass spectrometry and CRISPR screening, they revealed an essential and novel role of ORF29 in promoting the expression of late viral genes. The authors further confirmed these observations by genetic mutation and deletion of ORF29. The authors' conclusions are supported by the robust results that included proper controls and several different approaches.

Reviewer #3: In this manuscript, McCollum et al explore the role of the KSHV late transcription complex. They first use proximity labeling of the vTA complex and in QC experiments find that only ORF18 and ORF30 can tolerate the fusion with TurboID. Because of genomic constraints, they choose ORF18 as the gene to target in the virus generating a TurboID fusion BAC. Proximity labeling at 48h post infection led to the identification of 45 proteins that passed their criteria for true interactors with many belonging to GO groups including RNA processing/stability and this list included all known ORF18 interactors. A CRISPR screen was then used to define which of the interactors were important for late gene expression. Most cell genes were not, though XRN1, CASK and PABPC1 and PABPC4 co-depletion did suppress late gene expression. On the viral side, all 6 vTA components suppressed late gene expression and surprisingly ORF29, the KSHV terminase, was also important for this function. More detailed analysis of ORF29 expression found that it had expression kinetics between early and late genes. An ORF29 deleted virus was generated and this was incapable of generating infectious progeny, as would be expected given the essential role of herpesviral terminase proteins. However, in studies of early and late gene expression there was a surprising defect in both RNA levels of some early genes at 48 and72h post infection (ORF66), only at 48h (ORF66, ORF6-though not sign, and ORF68) and late gene RNA levels (K8.1). There was an apparent discordance between the protein levels of early gene products and the RNA (although protein was only measured at 72h in 4C). Late proteins ORF26 and K8.1 were clearly reduced in the ORF29 deleted virus at 72hpi. Mutational analysis then defined the Walker A/B and catalytic activity as required for the late gene expression phenotype in the absence of ORF29.

Overall this is an elegant, well controlled study that identifies a new interaction between the vTA and terminase complex suggesting a role for terminase in late gene expression. The hypothesis put forward in the model linking torsional stress to gene regulation is exciting and plausible.

**Part II – Major Issues: Key Experiments Required for Acceptance**

Reviewer #1: None

Reviewer #2: (No Response)

Reviewer #3: Only minor comments would be to repeat the western blots on a longer time course for early gene products in the KO and MR strains to clarify whether the expression defect of early genes is consistent or discordant between RNA and protein. What might the potential mechanism for early gene expression defects be at the 48h time point?

**Part III – Minor Issues: Editorial and Data Presentation Modifications**

Reviewer #1: 1) Lines 104-110. The assay described here (Fig S1B), unlike the excellent one used in Figure 2, should not be referred to as a "late" gene reporter assay. It is well established that authentic late gene transcription occurs in viral replication compartments (PMID: 23552415), even in alphaherpesviruses (PMID: 21555562) and requires continuous viral DNA replication (PMID: 29813138). The HEK293 cells used in this assay (as described in lines 652-663) harbor neither KSHV nor viral replication compartments and therefore cannot authentically report late gene transcription. I accept that, for reasons that are not entirely clear, this assay can distinguish between functional and non-functional vTA fusion proteins - probably via a transcription mechanisms that more closely approximates chromatinized transcription. Moreover, this assay is not required to established that ORF18-TurboID works as this is unequivocally established by Figs S1E and Fig1B. The risk of legitimizing this "late" reporter assay is that less sophisticated investigators could then use it to show that late gene transcription can occur in the absence of ORF29 (or even ORF9).

2) ORF31 is shown as interacting with ORF18 (Fig 1A), yet ORF31 is not shown among the KSHV proteins pulled down by ORF18-TurboID (Fig 1D) despite the claim that all of the known "ORF18-interacting proteins" were identified in the assay. Either the dotted line should be removed from Fig 1A or ORF31 added to Fig 1D.

3) I feel like the title is misleading given that ORF29 deletion impairs both early and late mRNAs. I agree with the authors that the selective effect on late proteins is likely due to translation of early mRNAs into protein prior to the ORF29 effect. However, the title implies ORF29's role in late gene transcription is on par with the vTAs.

Reviewer #2: (No Response)

Reviewer #3: (No Response)

PLOS authors have the option to publish the peer review history of their article (what does this mean?). If published, this will include your full peer review and any attached files.

Reviewer #1: No

Reviewer #2: No

Reviewer #3: **Yes: **Micah Luftig

Figure Files:

Data Requirements:

Reproducibility:

References:

---

## [Editor Report · Decision Letter 1]

27 Mar 2023

Dear Dr. Glaunsinger,

We are pleased to inform you that your manuscript 'The viral packaging motor potentiates Kaposi's sarcoma-associated herpesvirus gene expression late in infection' has been provisionally accepted for publication in PLOS Pathogens.

Best regards,

Sankar Swaminathan, MD

Academic Editor

PLOS Pathogens

Patrick Hearing

Section Editor

PLOS Pathogens

Kasturi Haldar

Editor-in-Chief

PLOS Pathogens

orcid.org/0000-0001-5065-158X

Michael Malim

Editor-in-Chief

PLOS Pathogens

orcid.org/0000-0002-7699-2064
---

## [Editor Report · Acceptance letter]

14 Apr 2023

Dear Dr. Glaunsinger,

We are delighted to inform you that your manuscript, "The viral packaging motor potentiates Kaposi's sarcoma-associated herpesvirus gene expression late in infection," has been formally accepted for publication in PLOS Pathogens.

Best regards,

Kasturi Haldar

Editor-in-Chief

PLOS Pathogens

orcid.org/0000-0001-5065-158X

Michael Malim

Editor-in-Chief

PLOS Pathogens

orcid.org/0000-0002-7699-2064